# Energy-Efficient AP Selection Using Intelligent Access Point System to Increase the Lifespan of IoT Devices

**DOI:** 10.3390/s23115197

**Published:** 2023-05-30

**Authors:** Seungjin Lee, Jaeeun Park, Hyungwoo Choi, Hyeontaek Oh

**Affiliations:** 1Institute for IT Convergence, Korea Advanced Institute of Science and Technology (KAIST), 291 Daehak-ro, Yuseong-gu, Daejeon 34141, Republic of Korea; dozze23@kaist.ac.kr (S.L.); hyeontaek@kaist.ac.kr (H.O.); 2School of Electrical Engineering, Korea Advanced Institute of Science and Technology (KAIST), 291 Daehak-ro, Yuseong-gu, Daejeon 34141, Republic of Korea; neojoey@kaist.ac.kr

**Keywords:** AP selection, energy efficiency, latency, internet of things, reinforcement learning

## Abstract

With the emergence of various Internet of Things (IoT) technologies, energy-saving schemes for IoT devices have been rapidly developed. To enhance the energy efficiency of IoT devices in crowded environments with multiple overlapping cells, the selection of access points (APs) for IoT devices should consider energy conservation by reducing unnecessary packet transmission activities caused by collisions. Therefore, in this paper, we present a novel energy-efficient AP selection scheme using reinforcement learning to address the problem of unbalanced load that arises from biased AP connections. Our proposed method utilizes the Energy and Latency Reinforcement Learning (EL-RL) model for energy-efficient AP selection that takes into account the average energy consumption and the average latency of IoT devices. In the EL-RL model, we analyze the collision probability in Wi-Fi networks to reduce the number of retransmissions that induces more energy consumption and higher latency. According to the simulation, the proposed method achieves a maximum improvement of 53% in energy efficiency, 50% in uplink latency, and a 2.1-times longer expected lifespan of IoT devices compared to the conventional AP selection scheme.

## 1. Introduction

The Internet of Things (IoT) is transforming our lives and workplaces, presenting unparalleled opportunities to improve efficiency, reduce costs, enhance safety, and drive innovation across a broad range of industries and applications. From smart homes and cities to healthcare, transportation, and industrial automation, IoT is reshaping how we engage with the world. One promising application of IoT technology is the use of unmanned aerial vehicles (UAVs) for effective data collection, enabling real-time monitoring and analysis in various contexts [1,2]. In particular, the healthcare industry is poised to experience significant economic growth worldwide by 2025, with estimates projecting annual growth between USD 1.1 trillion and 2.5 trillion due to the adoption and integration of IoT technology [3].

As the traffic on a Wi-Fi network increases, the cells covered by the network’s access point (AP) become smaller and more crowded. As a result, mobile terminals (MTs), including IoT devices, are present within multiple overlapping cells in Wi-Fi networks [4]. In this scenario, MTs typically connect to the AP with the strongest signal, which can result in contention and packet collisions during transmission due to the concentration of the devices on particular APs. Consequently, these repetitive transmissions can disrupt energy efficiency and increase latency at the device. Additionally, non-crowded APs are underutilized, which leads to lower overall network performance. Therefore, it is important to address the issue of selecting the optimal AP that considers IoT devices’ energy efficiency and latency in a multi-coverage Wi-Fi network environment.

There are two types of AP selection schemes in Wi-Fi networks: distributed and centralized. In a traditional distributed scheme, an MT selects an AP based on the received signal strength indication (RSSI) values between the MT and several available APs [5]. However, biased AP connections can occur when many MTs want to connect to a particular AP, which leads to load imbalance and poor quality of service (QoS) for MTs, including low throughput and latency performance [6]. Some studies have attempted to solve this problem by using the combination of RSSI values and other parameters [7,8], but distributed AP selection methods have limitations in addressing load balancing due to the limited information that MTs can obtain [9,10,11].

To address these issues, centralized AP selection methods have been proposed [12,13,14]. The centralized approach for AP selection involves the AP choosing the most suitable AP based on factors such as RSSI value and achievable throughput. This method can help to reduce the problem of unbalanced load and enhance network performance. However, this approach does not take into account the uplink traffic and the energy efficiency of IoT devices. When aiming to provide IoT services, it is crucial to consider the uplink traffic and the energy efficiency of IoT devices because the performance (e.g., reliability, durability, etc.) highly depends on the transmission activity of the IoT devices. For example, in healthcare IoT services, uplink traffic, including sensed IoT data, is frequently transmitted to the server, and the amount of uplink traffic is much more significant than that of downlink traffic. Therefore, rather than considering downlink traffic, the consideration of uplink traffic is more important. In addition, the frequent replacements of IoT devices due to the limited battery capacity is the most significant challenge for implementing good quality IoT service.

To solve the problems mentioned above, in [15] (our previous study on the iAP system), we proposed the iAP system that increases the energy efficiency of IoT devices when transmitting uplink IoT data after the AP connection procedure. However, we have recognized that the procedures for the initial AP selection and connection establishment also cause a large energy consumption of IoT devices, especially in crowded network environments. Such real-time connection dynamics between MTs and APs occur without the knowledge of future connections. The selection of an AP has a significant impact on network performance, specifically in terms energy efficiency, as it is influenced by factors such as the uplink traffic of APs and the distance between APs and their connected MTs. However, relying solely on the received signal strength indicator (RSSI) between the MT and AP is inadequate for achieving optimal connections. Moreover, the number of possible cases for connections between MTs and APs grows exponentially with the number of MTs, resulting in a large search space. To effectively explore this space while considering the influence of the current AP selection on future network performance, the adoption of a reinforcement learning algorithm is essential.

Therefore, in this paper, we propose an energy-efficient AP connection method using an intelligent AP (iAP) system [15] to increase the lifespan of IoT devices; particularly, we focus on an AP selection and connection method before transmitting uplink IoT data to achieve much better energy efficiency for IoT devices.

The main contributions of this paper are as follows:This paper proposes a novel energy-efficient AP selection scheme to increase the lifespan of IoT devices. To achieve this, we design an AP control system architecture that selects the optimal AP and controls operating parameters.We propose a new Energy and Latency Reinforcement Learning (EL-RL) model for optimal AP selection. The EL-RL model utilizes RSSI values and the number of connected IoT devices as input sequences for the AI model, with the aim of addressing the load-unbalancing problem and enhancing the energy efficiency of IoT devices. To the best of our knowledge, this represents the first attempt to consider the real-time connection dynamics and energy efficiency of IoT devices in the context of optimal AP selection.Based on the newly defined collision probability considering the retransmission of IoT devices, we design the energy consumption and latency estimation model of the overall IoT devices in Wi-Fi networks.We also analyzed the energy consumption and latency of IoT devices using a proposed energy-efficient AP selection scheme with an EL-RL model. Through extensive simulations, the proposed scheme achieved significant improvements, including a maximum of 53% in energy efficiency, 50% in uplink latency, and a 2.1-times improvement in the expected lifespan of IoT devices, compared to legacy AP selection schemes.

## 2. Related Works

Enhancing the energy efficiency of IoT devices is of paramount importance as it enables the provision of a diverse range of IoT services while simultaneously minimizing energy consumption. Significant research efforts have been devoted to this area, as evidenced by notable studies [16,17,18,19,20]. These works have made significant contributions to the understanding and development of energy-efficient solutions for IoT devices, offering valuable insights and strategies for improving their performance in terms of energy consumption and sustainability.

When multiple access points are overlapped, the selection of an appropriate AP becomes a critical concern. An energy-efficient AP selection method is required to address this challenge and enhance the energy efficiency and QoS for IoT devices in IoT services. As a result, numerous studies have focused on investigating AP selection schemes in both decentralized and centralized approaches [5,6,7,8,9,10,11,12,13,14]. These research endeavors aim to provide effective solutions for optimizing AP selection and improving the overall throughput and QoS of IoT devices in diverse IoT services (Table 1).

In legacy distributed AP selection schemes, the MT selects the AP with the strongest signal [5,9], which causes an unbalanced load across the network. In [9], the authors used an RSSI interval overlap degree determination method to improve positioning accuracy, but it did not address the load-unbalancing problem. Other AP selection schemes that utilize RSSI value and achievable throughput parameters also have limitations in AP load balancing and network utilization [7,11]. While in [7] the authors used a multi-armed bandits algorithm to enhance downlink throughput, it did not consider uplink traffic. In [11], the authors increased downlink throughput using RSSI value and achievable throughput, but the authors did not consider uplink traffic and energy consumption of MTs. Even centralized AP selection approaches primarily focus on downlink throughput [12,13], without considering uplink performance and collision probability. In [12], the authors used RSSI value and achievable throughput to select the optimal AP using a centralized approach, but the authors did not consider uplink traffic and energy efficiency. In [13], the authors used estimated RSSI values, which are obtained by a long short-term memory (LSTM) algorithm to improve positioning accuracy while reducing computational load and enhancing noise robustness, but the authors did not consider uplink traffic and energy efficiency. In general, AP selection studies have mainly emphasized increasing downlink performance rather than considering the uplink traffic and energy efficiency of IoT devices.

For more robust and durable IoT services, new AP selection proposals are necessary because IoT devices, which are the main component of the service, are sensitive to energy and uplink delay [15]. Therefore, a new energy-efficient AP selection scheme is required to overcome the problem of biased connection to a particular AP, which increases the collision probability of the network. Particularly, the biased connection can result in an increased amount of retransmissions at IoT devices, leading to higher energy consumption and uplink latency. Therefore, in this paper, we propose a new method that considers such problems to improve the performance of Wi-Fi networks.

## 3. System Description

In this section, we introduce a novel intelligent access point (iAP) control system for energy-efficient AP selection in uplink environments for IoT services. The proposed iAP control system is an advanced centralized AP selection scheme that considers the energy consumption and latency of IoT devices. The legacy AP selection scheme chooses the closest AP based on the highest RSSI value, which is not the best AP selection for the energy efficiency of IoT devices, as it causes the retransmission problem due to load unbalancing. In contrast, the proposed iAP control system selects the optimal AP by using not only the RSSI values of the IoT device but also the number of IoT devices in the AP coverage as input sequences for reinforcement learning. Additionally, the proposed iAP control system addresses the collision issue based on the formulation of collision probability considering the uplink transmissions of IoT devices aiming to minimize the number of collisions in the network. The proposed iAP control system solves the load-unbalancing problem and improves energy efficiency and uplink latency of IoT devices, as demonstrated by Figure 1, which shows an example of AP selection of IoT devices under overlapping APs. For example, from an IoT device perspective, the device can achieve more energy efficiency gain by balancing the energy consumption for uplink and retransmission. In other words, the IoT device may spend a little more energy to connect the sparse AP ($AP_{2}$) located far from the device, but it can significantly reduce retransmission energy consumption by avoiding collisions to connect the dense AP ($AP_{1}$) located closer from the device.

### 3.1. Architecture of iAP Control System

An overview of the proposed iAP control system is depicted in Figure 2. The software-based iAP controller is designed to facilitate energy-efficient AP selection for IoT devices. The iAP controller comprises the proposed Energy and Latency Reinforcement Learning (EL-RL) model, which is a reinforcement learning model that considers energy and latency factors to achieve optimal AP selection, as well as a transmission (Tx) power model and a location estimation model for better estimation and recommendation. The iAP controller interacts with the iAPs via an application programming interface (API) to ensure energy-efficient AP selection and load balancing. The selected iAP is responsible for managing the operational parameters of IoT devices to improve their energy efficiency.

The process of the proposed iAP control system for performing energy-efficient AP selection and deciding Tx power of IoT devices is illustrated in Figure 3. To begin, an IoT device initiates the process by transmitting a “probe request” message to iAPs. Upon receipt of the probe request message, the iAPs forward the message to the iAP controller along with the received RSSI value. Additionally, the iAPs periodically send local information, such as the number of connected IoT devices, to the iAP controller. The iAP controller utilizes global information, updated with the local information from the iAPs and various learning models, to select the optimal iAP and recommends the Tx power value for the IoT device. The selected iAP is then instructed to respond to the probe request with information on the recommended optimal transmitting power value of the IoT device. Upon receiving the probe response message from the selected optimal iAP, the IoT device establishes a connection with the optimal iAP and transmits IoT data with the recommended Tx power. The iAP controller employs an EL-RL model for optimal AP selection, which takes into account both energy and latency factors, to determine the energy-efficient AP selection. Additionally, the iAP controller employs a location estimation model to estimate the location of the IoT device and calculates the optimal transmit power value of the IoT device based on the estimated location.

### 3.2. Procedure of iAP Control System

The procedure of the iAP controller is presented in Figure 4. The summary of key symbol definitions is presented in Table 2 for reference. The iAP controller updates the global network status information that includes the number of IoT devices ($N_{IoT}$) connected to each iAP ($N_{AP}$) and the signal strength ($RSSI_{i,j}$) between the IoT device *i* and iAP *j*. Using the received $RSSI_{i,j}$ information from several iAPs, the iAP controller calculates the candidate iAP set $C_{i}$ based on the RSSI of IoT device *i* and the global network status. Then, the iAP controller employs the revised ideal CSMA (Carrier-Sense Multiple Access) network model to compute the energy consumption ($E_{i,j}$) and latency ($L_{i,j}$) of IoT device *i* in set $C_{i}$. Subsequently, the iAP controller trains the EL-RL model to optimize the objective function based on the average energy consumption and the average latency of IoT device *i* in the candidate iAP set $C_{i}$. Using the model, the iAP controller selects the iAP with the highest expected reward (considering the average energy consumption and average latency) for IoT device *i*. Moreover, the iAP controller determines the recommended transmitting power of IoT device *i* based on the fingerprinting map and assigns an iAP for the IoT device *i*, following which the iAP controller sends the control message to the selected iAP. Subsequently, the selected iAP transmits a “probe response” message to the IoT device *i*, which contains the recommended Tx power value. Upon receiving the “probe response” message, the IoT device *i* performs a connection handshake and transmits uplink data to the selected iAP with the recommended Tx power.

The functional architecture of the iAP system, consisting of IoT device, iAP, and iAP controller, is presented in Figure 5. Upon achieving the optimal iAP connectivity, the IoT devices wirelessly transmit IoT data to the iAP via the MQTT (Message Queuing Telemetry Transport) application layer using the TCP (Transmission Control Protocol) transmission method to ensure the protection and reliability of the data [15]. The iAP receives and stores the IoT data in its local cache before forwarding the data to the iAP controller, which is located on a cloud server and responsible for storing and analyzing the data in a database. Using the analyzed data, the iAP controller trains various AI models that are subsequently deployed to the iAP. The device energy management module in the iAP manages the energy consumption of the IoT devices by sending control messages to the IoT devices, which contain operating variable values. The IoT devices, in turn, adjust their data transmission period, DTIM (Delivery of Traffic Indication Map) value, Tx power, and other parameters based on the received control message, thus improving their energy efficiency.

## 4. Energy and Latency Reinforcement Learning (EL-RL) Model

The proposed Energy and Latency Reinforcement Learning (EL-RL) model is illustrated in Figure 6. The model is designed for iAP selection, where the environment sends state information in the form of $s_{t}$ to the EL-RL agent. The state $s_{t}$ is determined based on the RSSI between the IoT device and the candidate iAP, as well as the number of MTs currently connected to the iAP. At this stage, action $a_{t}$ represents the candidate iAP to connect the IoT device. The numerical solver then computes the reward $r_{t}$, taking into account the number of connected IoT devices and their distances from the chosen iAP. Additionally, the reward is calculated based on the average energy consumption and latency of IoT devices. Thus, the EL-RL model aims to minimize the average energy consumption and latency of all connected IoT devices, which is set as the objective function. The EL-RL agent receives the reward $r_{t}$ and selects a new action, and this process continues iteratively until the agent obtains the maximum reward through reinforcement learning. The notations used in the EL-RL model are defined as follows:

State, $s_{t}$–Global network information ($RSSI_{i,j}$, $n_{j}$) where *$i\in \left(1,2,...,N_{IoT}\right),\;j\in (1,2,...,$$N_{iAP}$*).–Candidate iAP set $C_{i}$ of the IoT device *i* where $C_{i}=\left(iAP_{1},iAP_{2},...,iAP_{cj}\right)$.Action, $a_{t}$–Select $iAP_{s}$ of the IoT device *i* where $C_{i}=\left(iAP_{1},iAP_{2},...,iAP_{cj}\right)$.–where subscript cj is the number of candidate iAPs for connecting the IoT device *i* among all iAPs.Reward (penalty), $r_{t}$–$\alpha \cdot \;E_{i,j}$ + $\beta \cdot \;L_{i,j}$.–where α is the weight for avg. energy consumption and β is the weight for avg. latency.Policy–Minimize the objective function $J\left(i\right)=argmin\left\{\alpha \cdot E_{avg}+\beta \cdot L_{avg}\right\}$.

In addition, the proposed iAP control system includes a location estimation ML model. This model employs a fingerprint method, which estimates location based on RSSI values by comparing them with reference point values stored in the database. The fingerprint method is widely recognized as the most suitable method for indoor positioning [21,22]. Once the location is estimated, the distances to each candidate iAP are calculated, and the recommended Tx power values are determined according to the adaptive Tx power equation (Equation (Equation 13)) in the Appendix A [15]. The iAP controller selects the optimal AP based on the EL-RL model and sends the recommended transmitting power to the IoT device. The iAP controller then updates the localization ML model and EL-RL model.

In the training procedure of the EL-RL model, each training data instance is obtained whenever a new connection is established between a MT and an AP. Each training data instance consists of the state $s_{t}$, which includes information such as the RSSI between the MT and AP and the number of already connected MTs for each AP. Additionally, it contains the action $a_{t}$ representing the selected AP for the connection and the reward $r_{t}$ associated with the chosen action in terms of network performance, such as latency and energy efficiency. To facilitate the training process, the training data is constantly stored in the iAP controller’s storage as new connections are made. From this dataset, a batch of training data is randomly selected for training the EL-RL model. This random selection helps ensure a diverse and representative sample of the training instances. To further enhance the learning process, the reward for each action in the selected training data is adjusted using the Proximal Policy Optimization (PPO) algorithm [23]. By adjusting the rewards, the model can better estimate the impact of each action on future network performance. During each epoch of training, the model’s parameters are iteratively updated using randomly chosen training data. This iterative process allows the model to gradually improve its performance and adapt to various network conditions. The training continues for several epochs until the total reward converges, indicating that the model has learned an optimal policy for AP selection.

### 4.1. Collision Probability

The energy consumption caused by traffic retransmissions resulting from packet collisions is demonstrated in Figure 7. When an IoT device and any other IoT devices try to simultaneously transmit a packet during the first transmission attempt from the IoT device perspective, a collision occurs between the transmitted packets, and a timeout for the IoT device occurs because an ACK(Acknowledgement) packet has not been received. Once the channel becomes idle again, the IoT device attempts a second transmission using a random backoff time within the double contention window size. The same process applies to the collisions encountered during the second through sixth transmission attempts. If a collision happens even on the seventh transmission attempt, the packet is discarded, and there is no further retransmission attempt.

To examine the energy consumption attributed to retransmissions, we conduct mathematical calculations of collision probabilities based on realistic collision simulations. As per the IEEE 802.11 standardization, we consider that the IoT device could transmit the same packet a total of seven times, which includes the initial transmission attempt. Hence, the maximum number of retransmission attempts (*m*) is six [24,25], the minimum contention window size $CW_{min}$ is 31 time slots, the maximum contention window size $CW_{max}$ is 1023 time slots, and the maximum number of recursive attempts to increase CW is 6 [24,25]. We define the collision probability for each transmission attempt as $P_{c}\left(n\right)$ and the transmission attempt probability as $P_{a}\left(n\right)$ in follows.

$P_{c}\left(n\right)$: Collision probability of the *n*th transmission attempt–$P_{c}\left(1\right)$: Collision probability of the 1st transmission attempt.–$P_{c}\left(2\right)$: Collision probability of the 2nd transmission attempt after 1st transmission failure.–$P_{c}\left(3\right)$: Collision probability of the 3rd transmission attempt after 2nd transmission failure.–

⋮


⋮

–$P_{c}\left(7\right)$: Collision probability of the 7th transmission attempt after 6th transmission failure.$P_{a}\left(n\right)$: Transmission probability of the *n*th transmission attempt–$P_{a}\left(1\right)=\left(1-e^{-\lambda }\right)$: Transmission probability of 1st transmission attempt.–$P_{a}\left(2\right)=\left(1-e^{-\lambda }\right)P_{c}\left(1\right)$: Transmission probability of the 2nd transmission attempt after 1st transmission failure.–$P_{a}\left(3\right)=\left(1-e^{-\lambda }\right)P_{c}\left(1\right)P_{c}\left(2\right)$: Transmission probability of the 3rd transmission attempt after 1st and 2nd transmission failure.–

⋮


⋮

–$P_{a}\left(7\right)=\left(1-e^{-\lambda }\right)P_{c}\left(1\right)P_{c}\left(2\right)P_{c}\left(3\right)P_{c}\left(4\right)P_{c}\left(5\right)P_{c}\left(6\right)$: Transmission probability of the 7th transmission attempt after 6th transmission failure.

Therefore, the transmission probability of the *n*th transmission attempt, $P_{a}\left(n\right)$ is given by Equation (Equation 1),
(1)$$P_{a}\left(n\right)=\left(1-e^{-\lambda }\right)\prod_{1}^{n}P_{c}\left(n-1\right),\;s.t.\;P_{c}\left(0\right)=1.$$

In this paper, a collision occurs when more than one IoT devices share the same time slot for attempting uplink transmissions. For example, when one device among *N* devices is trying to transmit within a certain time slot, another device among N−1 devices may try to transmit simultaneously. We take into account the concurrent transmission attempt in the following collision model. The transmission collision probability is formulated from a new perspective in Equation (Equation 2),
(2)$$\begin{array}{cc} P_{c}\left(n\right) & =\sum_{a_{1},a_{2},a_{3},a_{4},a_{5},a_{6},a_{7}}P_{s}(\begin{array}{c} N-1\\ a_{1} \end{array})P_{a}{\left(1\right)}^{a_{1}}(\begin{array}{c} N-1-a_{1}\\ a_{2} \end{array})P_{a}{\left(2\right)}^{a_{2}}\ldots (\begin{array}{c} N-1-(a_{1}+a_{2}+\ldots +a_{6})\\ a_{7} \end{array})P_{a}{\left(7\right)}^{a_{7}}{\left(1-P_{A}\right)}^{a_{0}}\\ \; & =\sum_{a_{1},a_{2},a_{3},a_{4},a_{5},a_{6},a_{7}}P_{s}\frac{(N-1)!}{a_{1}!a_{2}!a_{3}!a_{4}!a_{5}!a_{6}!a_{7}!a_{0}!}P_{a}{\left(1\right)}^{a_{1}}P_{a}{\left(2\right)}^{a_{2}}\ldots P_{a}{\left(7\right)}^{a_{7}}{\left(1-P_{A}\right)}^{a_{0}}\\ \; & =\sum_{a_{1},a_{2},a_{3},a_{4},a_{5},a_{6},a_{7}}P_{s}\frac{(N-1)!}{\prod_{n=0}^{m+1}\left(a_{n}\right)!}\prod_{n=1}^{m+1}P_{a}{\left(n\right)}^{a_{n}}{\left(1-P_{A}\right)}^{a_{0}},\;s.t.N-1=\sum_{n=0}^{m+1}a_{n}, \end{array}$$

–where $P_{s}=\left\{\begin{array}{c} 1,\mathrm{when}\mathrm{there}\mathrm{is}\mathrm{another}\mathrm{transmission},\\ 0,\mathrm{when}\mathrm{there}\mathrm{is}\mathrm{no}\mathrm{other}\mathrm{transmission}, \end{array}\right.$–where $N-1=\sum_{n=0}^{m+1}a_{n}$–where $P_{A}=\sum_{n=1}^{m+1}P_{a}\left(n\right)$

This collision probability is calculated by considering the packet collision probability within a single arbitrary time slot. Additionally, this collision probability considers the actual collision probability for ML, which can be solved numerically. Concerning the transmitting devices at an arbitrary time slot, the number of devices attempting the first transmission in that time slot is represented by $a_{1}$, and the number of devices attempting the second transmission is expressed as $a_{2}$. Likewise, $a_{n}$ describes the number of devices attempting *n*th transmission in that time slot for n=0,1,2,...,7. In addition, $a_{0}$ is the number of devices with no transmission attempt in the same time slot.

The collision probability is defined as the sum of the values multiplied by the number of cases in which collision can happen and the transmission attempt probability. Here, if there is no transmission from any device at that time slot, the $P_{s}$ has a value of 0, and it is considered that no collision has occurred. Furthermore, $P_{A}$ is defined as the sum of all attempted transmission probabilities. The collision probability based on these actual collisions was calculated with numerical techniques.

### 4.2. Energy and Latency of IoT Devices

In this subsection, we present the average energy consumption and the average latency model of IoT devices based on the collision probability. The average energy consumption of IoT devices can be obtained as the sum of the product of the probability of all transmission attempts, the probability of successful transmission without collision, and the energy consumption value according to the *n*th transmission attempt. The average energy consumption of IoT devices is given as Equation (Equation 3),
(3)$$E_{avg}=\sum_{1}^{n}\frac{P_{a}\left(n\right)}{1-e^{-\lambda }}\left[1-P_{c}\left(n\right)\right]E\left(n\right).$$

The energy consumed by the *n*th transmission attempt is the sum of the product of the operation time of each operation mode and the power used in that operation mode as follows in Equation (Equation 4).
(4)$$E\left(n\right)=P_{tx}^{adaptive}\cdot T_{tx}\left(n\right)+P_{rx}\cdot T_{rx}\left(n\right)+P_{sleep}\cdot T_{sleep}\left(n\right).$$

The total Tx mode time according to the transmission attempt consists of data transmission time and ACK transmission time in Equation (Equation 5). The data transmission time can be obtained by multiplying the number of transmission attempts by the time required to send one transmission data, and the ACK transmission time can be obtained by the time required to send an L2ACK message once.
(5)$$T_{tx}\left(n\right)=n\cdot \frac{N_{data}}{Blog_{2}\left(1+\gamma \right)}+\frac{N_{L2ack}}{Blog_{2}\left(1+\gamma \right)},$$
where $N_{data}$ is a 104 bytes, $N_{L2ack}$ is a 54 bytes, *B* is a 160 kHz, and γ is a 40 dB [15].

The total Rx (Receive) mode time according to the transmission attempt is given by
(6)$$T_{rx}\left(n\right)=\left(n-1\right)\cdot T_{ACKtimeout}+T_{ACKtime}+T_{beacons},$$
where $T_{ACKtimeout}$ is a 337 μs, $T_{ACKtime}$ is a 44 μs [12], and $T_{beacons}$ is a $\left|\frac{I_{period}}{n_{dtim}\cdot I_{beacon}}\right|\cdot t_{beacon}$
μs [15]. The time calculated in Rx mode is the sum of the ACKtime time value, the beacon reception time value, and the product of the number of times sent so far and the time set by ACKtimeout.

The total sleep mode time according to the transmission attempt is given by
(7)$$T_{sleep}\left(n\right)=I_{period}-T_{tx}\left(n\right)-T_{rx}\left(n\right),$$
where $I_{period}$ is a 1 s of the transmission period. The total sleep mode time per transmission attempt can be obtained by subtracting the Tx mode time and the Rx mode time from the period.

In addition, the adaptive Tx power according to the distance, $P_{tx}^{adaptive}$, can be obtained as Equation (Equation 13) in the Appendix A [15]. The average uplink latency of IoT devices is calculated by the below Equation (Equation 8). The average latency is composed of the average backoff time, the average transmission time for successful delivery, and the average collision time for transmission failure according to the *n*th transmission attempts,
(8)$$L_{avg}=\sum_{1}^{n}\frac{P_{a}\left(n\right)}{1-e^{-\lambda }}\left[\sigma \left(n\right)+\left(1-P_{c}\left(n\right)\right)T_{a}\left(n\right)+P_{c}\left(n\right)T_{c}\left(n\right)\right].$$

The average backoff time of *n*th transmission attempts is given by
(9)$$\sigma \left(n\right)=\frac{2^{n-1}\cdot W_{initial}}{2}\cdot timeslot,$$
where timeslot is a 20 μs, and $W_{initial}$ is a 16 as a default value [12]. The average transmission time for the successful delivery of *n*th transmission attempts is given by
(10)$$T_{a}\left(n\right)=\left(n-1\right)T_{c}\left(n\right)+\frac{N_{data}}{Blog_{2}\left(1+\gamma \right)}+SIFS+T_{ACKtime}+DIFS,$$
where SIFS is a 10 μs, $T_{ACKtime}$ is a 44 μs, and DIFS is a 50 μs [12]. The average collision time for transmission failure of *n*th transmission attempts is given by
(11)$$T_{c}\left(n\right)=n\cdot \left(\frac{N_{data}}{Blog_{2}\left(1+\gamma \right)}+T_{ACKtimeout}+DIFS\right),$$
where $T_{ACKtimeout}$ is a 337 μs, and DIFS is a 50 μs [12].

To calculate the average consumed energy of an IoT device, we use the RSSI values and the number of IoT devices that are connected to the iAPs. Moreover, to calculate the average latency of an IoT device, we use the number of IoT devices that are connected with iAPs for load balancing. The objective function of the proposed EL-RL model has defined below in Equation (Equation 12),
(12)$$J\left(i\right)=argmin\left\{\alpha \cdot E_{avg}+\beta \cdot L_{avg}\right\},$$
where α and β are the weight of average energy consumption and average latency, respectively. The goal of the objective function is to minimize the weighted sum of the average energy value and average latency value.

## 5. Performance Evaluation

The simulator uses Python and the PyTorch library for the PPO algorithm implementation [26]. The parameter settings for simulation are shown in Table 3.

For the simulation, we assume the total number of APs is three, the distance between the APs is 20 m, and the cell coverage is 15 m. In addition, it is assumed that the IoT devices in each APs are normally distributed with respect to the iAP location, which is placed at the center of the cell. The distribution ratio of IoT between APs are assumed to be [1:1:1], [1:9:9], and [1:10:3]. These represent hotspot scenarios: balanced scenario, two hotspots scenario, and one hotspot scenario, respectively. The total numbers of IoT devices applied to the simulation are 50, 100, 150, and 200. Reinforcement learning of the EL-RL model is performed based on the PPO algorithm, which shows the best performance and fastest learning in various fields [27] (Figure 8).

The reason for using the PPO algorithm is as follows. First, it is rare for a sequence to produce a similar state because the state in a sequence is defined by the distance between the IoT device and the AP and the number of devices connected to the AP. Second, in order to train EL-RL model from numerous amount of various sequences, we must carefully consider the effect of current actions on future actions, i.e., the final return value. Therefore, we implement an advantage actor-critic-based PPO algorithm as a value-based algorithm that can efficiently consider the return value for the current action.

The agent of the proposed EL-RL model is based on the PPO algorithm. The state, action, reward weight values, and epoch of the EL-RL model for the simulation are as follows.

State: $\left(RSSI\left(AP_{1}\right),RSSI\left(AP_{2}\right),RSSI\left(AP_{3}\right),N\left(AP_{1}\right),N\left(AP_{2}\right),N\left(AP_{3}\right)\right)$–$RSSI(AP_{n})$ represent the RSSI value of the $IoT_{x,y}$ at the $AP_{n}$.–$N(AP_{n})$ represent the number of connected IoT devices in the $AP_{n}$.Action: $\left\{(1,0,0),(0,1,0),(0,0,1)\right\}$–(1,0,0) represent selection to $AP_{1}$.–(0,1,0) represent selection to $AP_{2}$.–(0,0,1) represent selection to $AP_{3}$.Reward weight value: $-1000\times {\bar{E}}_{x,y}-1000\times {\bar{L}}_{x,y}$–${\bar{E}}_{x,y}$: Average energy consumption of $IoT_{x,y}$.–${\bar{L}}_{x,y}$: Average uplink latency of $IoT_{x,y}$.Epoch: 200

We compare three AP selection models for performance evaluation. First, the legacy AP selection model that only uses RSSI value to select AP is presented as ‘legacy AP with RSSI’. Second, the proposed iAP selection model that only uses RSSI value to select iAP with an adaptive Tx power is expressed as ‘proposed iAP with RSSI’. Last, the proposed iAP selection model that uses the EL-RL agent to select iAP is denoted as ‘proposed iAP with EL-RL’. With three AP selection models, we consider three cases regarding distribution ratios of IoT devices between APs as follows.

Case 1: refers to the balanced case (the distribution ratio between APs is [1:1:1]).Case 2: refers to the two hotspots case (the distribution ratio between APs is [1:9:9]).Case 3: refers to the one hotspot case (the distribution ratio between APs is [1:10:3]).

The results for each model in all experiments are the average value obtained from 500 simulations.

Figure 9 presents the average energy consumption of IoT devices according to the distribution ratio between APs. In all cases, the average energy consumption of IoT devices shows an increasing trend as the number of devices increases. For Case 1, the energy consumption performance of the two proposed iAP models (namely ‘proposed iAP with RSSI’ and ‘proposed iAP with EL-RL’) are better than that of the ‘legacy AP with RSSI’ model, but the energy consumption values of the two models are comparable as shown in Figure 9a. Since Case 1 is already load-balanced, it shows similar performance between the two proposed models. However, the two proposed iAP models demonstrate lower energy consumption, at 63∼66%, compared to the legacy AP model because of the adaptive Tx power and the prompt ACK reception function in the iAP system. In Case 2 and Case 3, as shown in Figure 9b,c, respectively, the average energy consumption of IoT devices increases with the increasing number of IoT devices, a similar trend to Case 1. In Case 2, where there are two hotspot APs, the two proposed iAP models exhibit energy consumption performance ranging from 62% to 65% compared to the legacy AP model. On the other hand, in Case 3, where there is only one hotspot AP, the two proposed iAP models demonstrate better performance in terms of energy consumption ranging from 47% to 64% compared to the legacy AP model. Especially the ‘proposed iAP with EL-RL’ model performs the best in Case 3, exhibiting energy consumption of only 47.1% compared to the ‘legacy AP with RSSI’ model, with a total of 100 IoT devices. This is because the ‘proposed iAP with EL-RL’ model has a better load-balancing effect that reduces retransmission energy.

Figure 10 displays the average energy consumption of IoT devices for cases with respect to the different numbers of IoT devices. The results indicate that the two proposed iAP models outperform the legacy AP model in terms of energy consumption performance. Particularly, the ‘proposed iAP with EL-RL’ model demonstrates the best energy consumption performance, achieving an energy reduction of 47.1% in the 1:10:3 distribution of 100 IoT devices. This outcome is due to the ‘proposed iAP with EL-RL’ model’s load-balancing scheme, which selects the optimal AP while taking into account both energy consumption and latency.

Figure 11 presents the average uplink latency of IoT devices according to the distribution ratio between APs. Figure 11a shows the average uplink latency of IoT devices for Case 1. The average uplink latency of each model increases as the number of IoT devices increases due to retransmissions resulting from packet collisions. However, the two proposed iAP models exhibit almost the same average uplink latency as the legacy AP model since the APs are already load-balanced. Figure 11b,c shows the average uplink latency of IoT devices for Case 2 and Case 3, respectively. In Case 2, where there are two hotspot APs, the ‘proposed iAP with EL-RL’ model demonstrates a latency ranging from 71% to 94% compared to the legacy AP model. This is because only the ‘proposed iAP with EL-RL’ model selects the AP, taking into account the latency of IoT devices. Furthermore, in Case 3, where there is only one hotspot AP, the ‘proposed iAP with EL-RL’ model exhibits better performance, with a latency ranging from 50% to 82% compared to the legacy AP model. From this, we can see that the ‘proposed iAP with EL-RL’ model shows better latency performance as the unbalanced load situation worsens.

The average uplink latency of IoT devices under each case with the different number of IoT devices is depicted in Figure 12. The results indicate that in Cases 2 and 3 where load balancing is required, the ‘proposed iAP with EL-RL’ model is superior in terms of latency performance to both the ‘legacy AP with RSSI’ model and the ‘proposed iAP with the RSSI’ model. This is because the EL-RL model minimizes the number of retransmissions through load balancing. Specifically, the ‘proposed iAP with EL-RL’ model demonstrates the best latency performance, achieving a latency reduction of 50.5% in the 1:10:3 distribution ratio of 100 IoT devices. This outcome is due to the ‘proposed iAP with EL-RL’ model’s load-balancing scheme, which chooses the optimal AP considering the latency of IoT devices.

Figure 13 presents the expected lifespan of an IoT device, under the different distribution ratios between APs. In Case 2 of the 1:9:9 distribution ratio between APs, the expected lifespan of an IoT device is shown in Figure 13a. The ‘proposed iAP with EL-RL’ model can significantly enhance the expected lifespan, with an improvement ranging from 1.6 times to 1.9 times roughly when compared to the ‘legacy AP with RSSI’ model. Furthermore, Figure 13b displays the expected lifespan of an IoT device according to Case 3 of the 1:10:3 distribution ratio between APs. The ‘proposed iAP with EL-RL’ model offers an even more significant improvement in the expected lifespan, roughly ranging from 1.7 times to 2.1 times when compared to the legacy AP model. From this, it can be seen that the ‘proposed iAP with EL-RL’ model shows better energy-saving performance as the unbalanced load situation deepens. As such, the increased expected lifespan of IoT devices using the ‘proposed iAP with EL-RL’ model can be of great help in providing various IoT services by solving the problem of frequent battery replacement.

The generalization of IoT device location (i.e., the location of each IoT device has continuously changed as epoch increased) in the EL-RL model is demonstrated in Figure 14 under three cases, each with a total of 100 devices and different distribution ratios between APs. In Case 1 where the distribution ratio is 1:1:1, Figure 14a displays the location generalization. Case 2 with a distribution ratio of 1:9:9 is presented in Figure 14b. Finally, Figure 14c illustrates Case 3 where the distribution ratio is 1:10:3. As the epoch progresses, the IoT devices located at the overlapping section tended to select the AP connected with a smaller number of IoT devices to maintain stable load balancing in terms of energy efficiency and latency. Therefore, regardless of the distribution of IoT devices, the proposed EL-RL model can be stably trained under the generalization of IoT device location, and improve the energy efficiency and latency performance of IoT devices.

The convergence analysis of the EL-RL model is demonstrated in Figure 15 under three different distribution ratios between APs, in order to examine its performance in various scenarios. Figure 15a depicts the convergence behavior of the model when the distribution ratio is 1:1:1. The reward of the EL-RL model quickly and efficiently converges after approximately 25 epochs of training, while the energy consumption and latency also gradually converge as the epochs progress. Similarly, the convergence behavior of the EL-RL model is presented in Figure 15b when the distribution ratio is 1:9:9. As observed in the previous case, the reward, energy consumption, and latency of the EL-RL model converge efficiently after approximately 25 epochs of training. Finally, Figure 15c illustrates the convergence behavior of the EL-RL model when the distribution ratio is 1:10:3. The reward of the model gradually converges as the epochs progress, indicating that the reinforcement learning was successful. Although the reward changes rapidly in some cases, the range of change decreases as the learning progresses and ultimately converges. Additionally, the energy consumption and latency of the EL-RL model also converge as the epochs progress.

To address the training and inference time of the EL-RL model, we provide comprehensive information in Table 4, which summarizes the average training time per epoch and the average inference time per input instance. The simulations were conducted on a computer system with a 64-bit Intel Core i7-800 CPU, and 16 GB of RAM. The simulation results reflect the duration required for training the EL-RL model and the inference time for making AP selections across different cases. It is noteworthy that the average training time per epoch increases with a higher number of MTs. Nevertheless, the overall training duration remains within 25 epochs, equivalent to less than 10 min. Furthermore, the training and inference procedures can be decoupled. The iAP operates using the most recently updated EL-RL model, which is redistributed to the iAPs when the model is updated at the iAP controller with an accumulated training dataset. This approach enables dynamic and iterative training, enhancing the model’s effectiveness over time. With an average inference time below 0.5 milliseconds, the EL-RL model has a minimal impact on the overall time required for establishing connections, typically measured in seconds [28]. Therefore, the EL-RL model demonstrates its feasibility for real-world AP selection scenarios without significantly increasing connection setup delays. Finally, while the training and inference of the EL-RL model primarily utilize the CPU, incorporating GPU acceleration can further reduce processing time in both the training and inference stages.

## 6. Conclusions

In this paper, we propose an energy-efficient AP selection scheme for IoT devices that uses reinforcement learning to minimize energy consumption and latency. To achieve this goal, we develop an iAP control system for selecting the optimal AP in Wi-Fi networks. We also introduce a novel energy-efficient AP selection model, EL-RL model, which utilizes RSSI values and the number of IoT devices connected to APs to balance the load. Additionally, we design an energy and latency reinforcement learning (EL-RL) model to address the load-unbalancing problem. Furthermore, we control the adaptive Tx power of IoT devices by employing a location estimation ML model and a Tx power recommendation model. We evaluate the proposed scheme by analyzing the energy consumption, uplink latency, and collision probability in Wi-Fi networks. Our results show that the proposed scheme can achieve a maximum improvement in energy efficiency of 53%, a 50% reduction in latency, and a 2.1-times improvement in the expected lifespan of IoT devices.

In future research, it would be valuable to explore the potential limitations and extensions of our proposed scheme. One possible direction is to investigate the applicability of the EL-RL model and iAP control system for different types of IoT devices or in diverse environmental conditions, such as an industrial IoT service. Additionally, the proposed scheme could be adapted to incorporate other relevant factors, such as network congestion or device mobility, to further optimize energy efficiency and latency. By addressing these aspects, we can continue to enhance the performance and versatility of the proposed scheme, making it more robust and adaptable for various IoT scenarios.

## Figures and Tables

**Figure 1 sensors-23-05197-f001:**
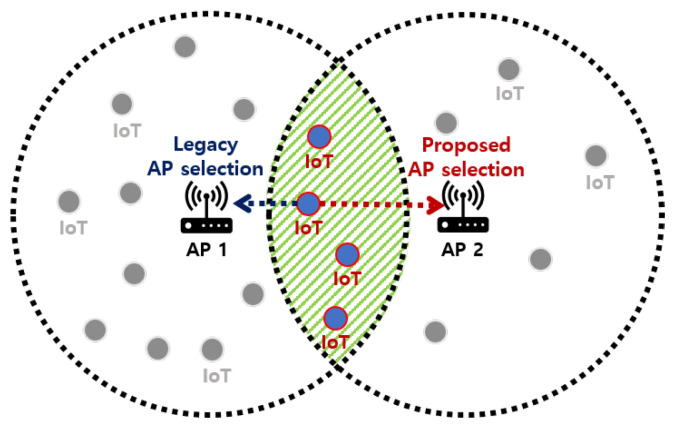
An example of AP selection of IoT devices.

**Figure 2 sensors-23-05197-f002:**
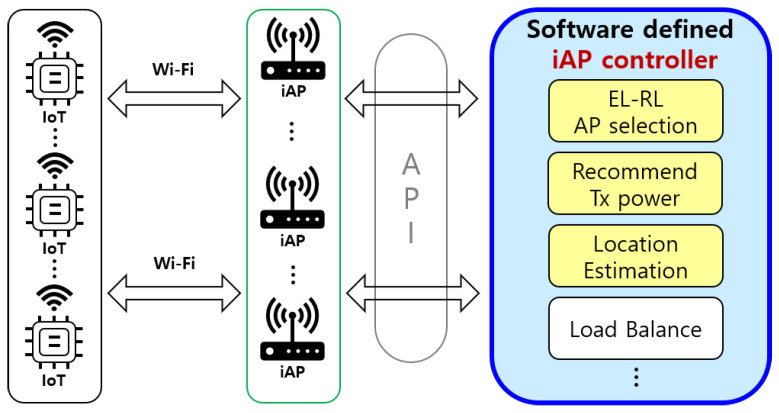
A overview of iAP control system.

**Figure 3 sensors-23-05197-f003:**
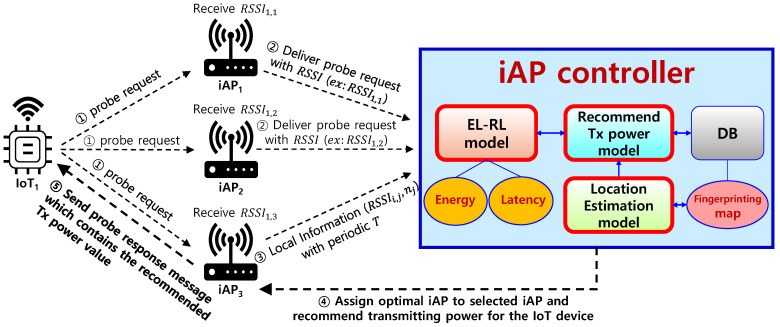
Process of iAP control system.

**Figure 4 sensors-23-05197-f004:**
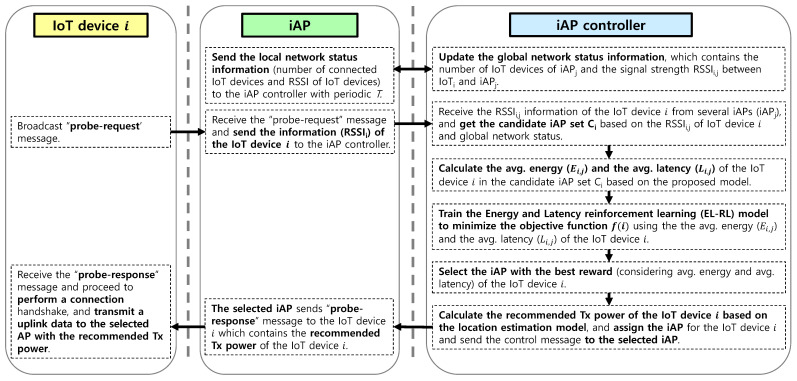
Procedure of iAP controller.

**Figure 5 sensors-23-05197-f005:**
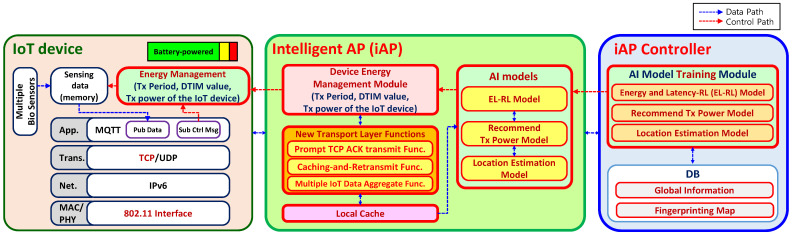
Functional architecture of iAP system.

**Figure 6 sensors-23-05197-f006:**
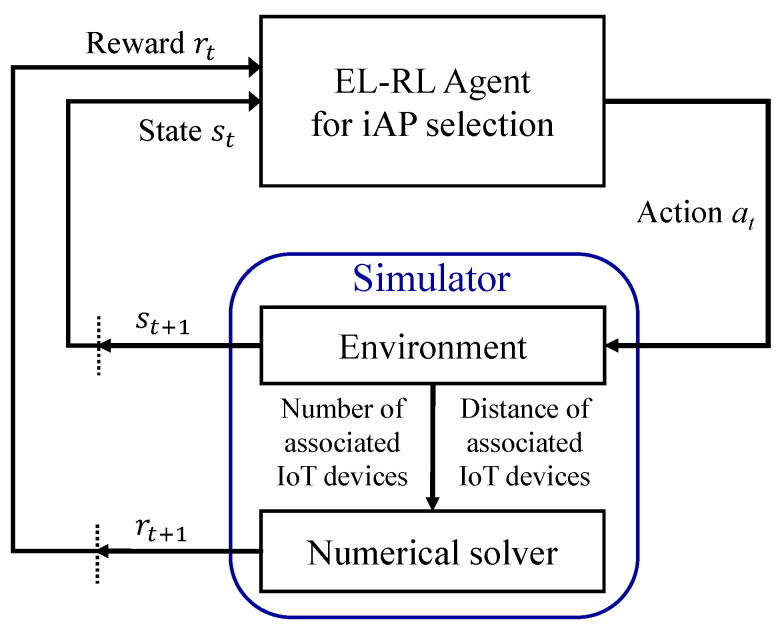
Energy and Latency Reinforcement Learning (EL-RL) model.

**Figure 7 sensors-23-05197-f007:**
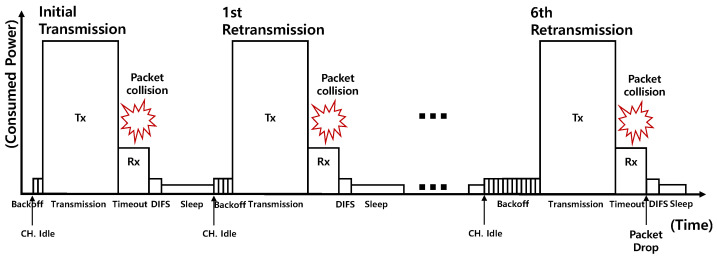
Energy consumption according to retransmissions.

**Figure 8 sensors-23-05197-f008:**
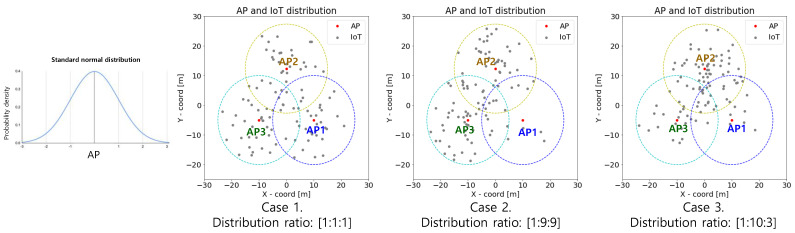
EL-RL model environment setting.

**Figure 9 sensors-23-05197-f009:**
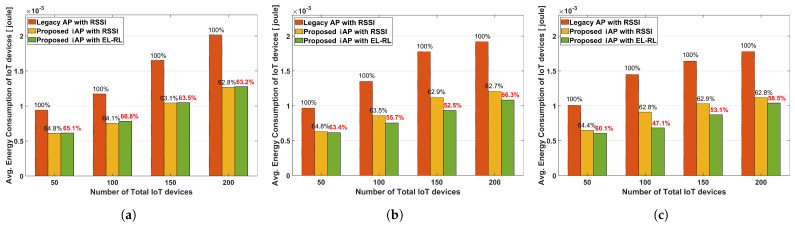
Average energy consumption of IoT devices according to the distribution ratio. (**a**) Case 1. distribution ratio 1:1:1. (**b**) Case 2. distribution ratio 1:9:9. (**c**) Case 3. distribution ratio 1:10:3.

**Figure 10 sensors-23-05197-f010:**
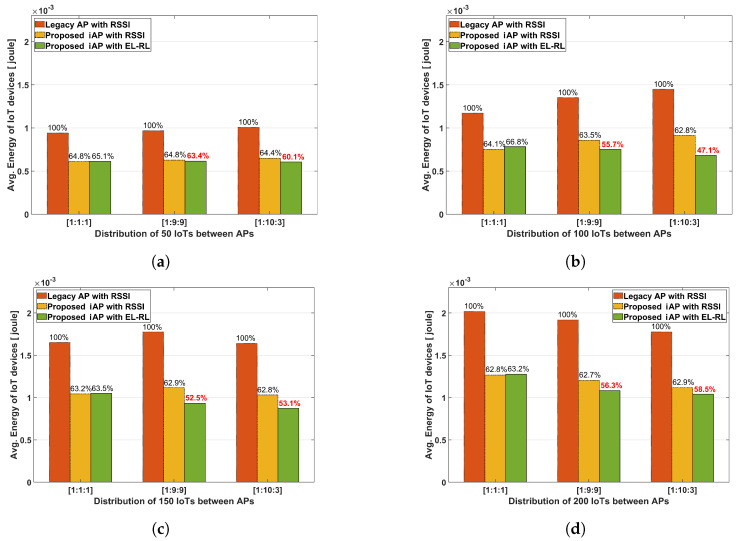
Average energy consumption of IoT devices according to density. (**a**) Total 50 IoT devices. (**b**) Total 100 IoT devices. (**c**) Total 150 IoT devices. (**d**) Total 200 IoT devices.

**Figure 11 sensors-23-05197-f011:**
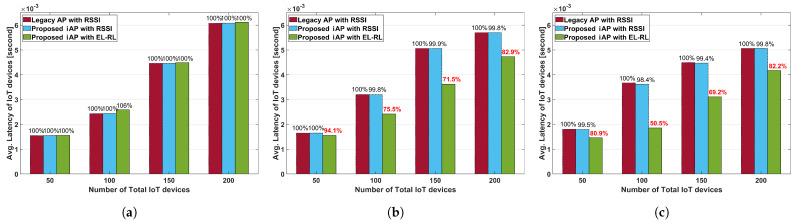
Average uplink latency of IoT devices according to the distribution ratio. (**a**) Case 1. distribution ratio 1:1:1. (**b**) Case 2. distribution ratio 1:9:9. (**c**) Case 3. distribution ratio 1:10:3.

**Figure 12 sensors-23-05197-f012:**
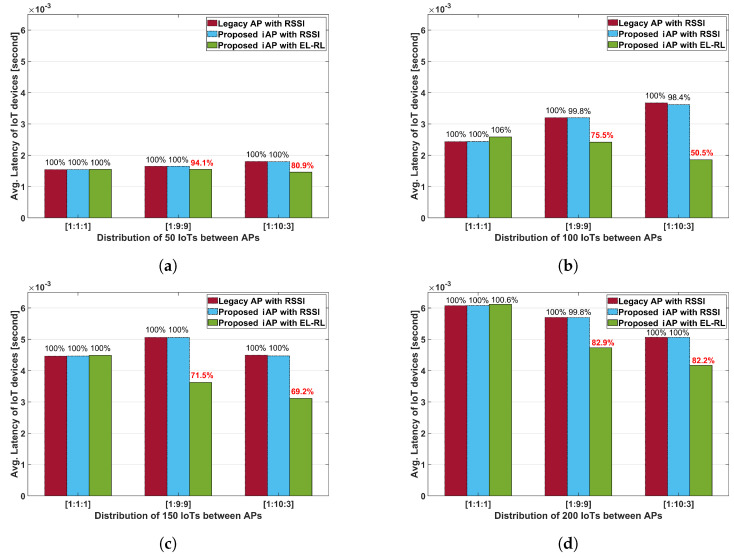
Average uplink latency of IoT devices according to the density. (**a**) Total 50 IoT devices. (**b**) Total 100 IoT devices. (**c**) Total 150 IoT devices. (**d**) Total 200 IoT devices.

**Figure 13 sensors-23-05197-f013:**
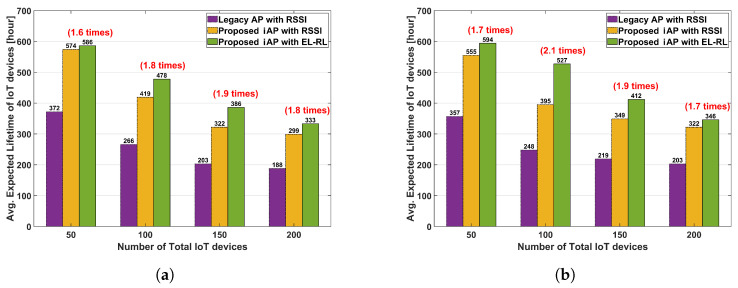
Expected lifespan of an IoT device according to the distribution ratio. (**a**) Case 2. distribution ratio 1:9:9. (**b**) Case 3. distribution ratio 1:10:3.

**Figure 14 sensors-23-05197-f014:**
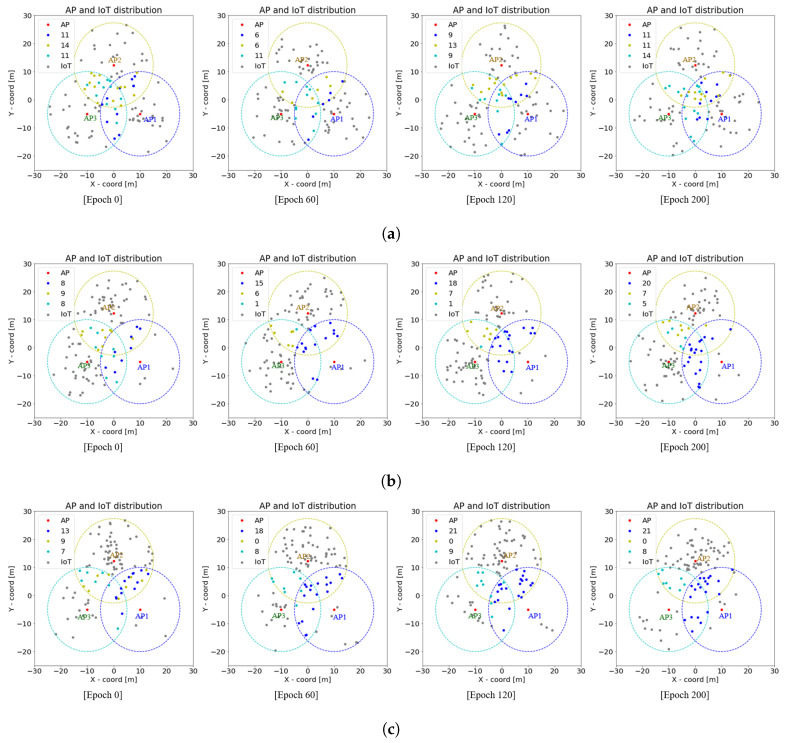
Generalization of the location of IoT devices in EL-RL model. (**a**) Generalization of Case 1: distribution ratio 1:1:1. (**b**) Generalization of Case 2: distribution ratio 1:9:9. (**c**) Generalization of Case 3: distribution ratio 1:10:3.

**Figure 15 sensors-23-05197-f015:**
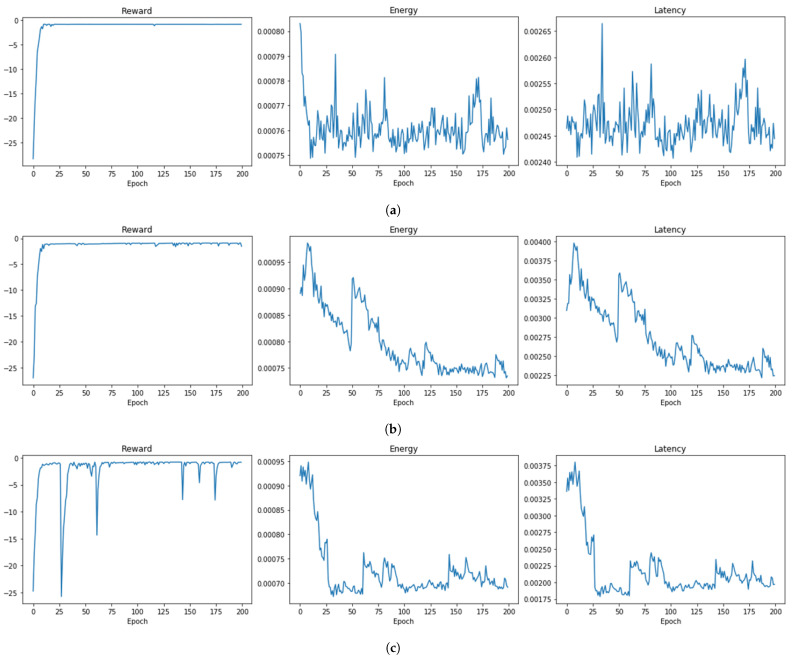
Convergence of EL-RL model. (**a**) Convergence of Case 1. distribution ratio 1:1:1. (**b**) Convergence of Case 2. distribution ratio 1:9:9. (**c**) Convergence of Case 3. distribution ratio 1:10:3.

**Table 1 sensors-23-05197-t001:** Comparison of related works.

Related Work	AP Selection Scheme	Description	Advantages	Disadvantages
[5,9]	Distributed	Use RSSI	Simple	Load unbalancing problem No consider energy consumption
[7]	Distributed	Use RSSI and multi-armed bandits	Enhance downlink throughput	Load unbalancing problem No consider energy consumption
[11]	Distributed	Use RSSI and achievable throughput	Enhance downlink throughput	Load unbalancing problem No consider energy consumption
[12]	Centralized	Use RSSI and achievable throughput	Enhance downlink throughput	No consider uplink traffic No consider energy consumption
[13]	Centralized	Use RSSI and LSTM	Enhance downlink throughput Reduce computational load	No consider uplink traffic No consider energy consumption
Proposed	Centralized	Use RSSI and number of IoTs	Enhance energy efficiency Reduce latency	-

**Table 2 sensors-23-05197-t002:** Key notations.

Notation	Definition	Notation	Definition
$N_{IoT}$	Number of IoT devices	$N_{AP}$	Number of iAP
$RSSI_{i,j}$	Signal strength between IoT *i* and iAP *j*	$C_{i}$	Candidate iAP set of IoT *i* for connection
$E_{i,j}$	Energy consumption of IoT *i* when connecting to iAP *j*	$L_{i,j}$	Uplink latency of IoT *i* when connecting to iAP *j*
$E_{avg}$	Average energy consumption of overall IoTs	$L_{avg}$	Average uplink latency of overall IoTs
$P_{c}\left(n\right)$	Collision prob. of the *n*th transmission attempt	$P_{a}\left(n\right)$	Transmission prob. of the *n*th transmission attempt
$a_{n}$	Number of IoTs attempting *n*th transmission attempt	$P_{A}$	Sum of all attempted transmission probs.
E(n)	Consumed energy by the *n*th transmission attempt	$P_{tx}^{adaptive}$	Adaptive Tx power of IoT device
$T_{tx}\left(n\right)$	Total Tx mode time of *n*th transmission attempt	$T_{rx}\left(n\right)$	Total Rx mode time of *n*th transmission attempt
$T_{sleep}\left(n\right)$	Total sleep mode time of *n*th transmission attempt	$P_{rx}$	Receive mode power of IoT devices
$P_{sleep}$	Sleep mode power of IoT devices	$N_{data}$	Amount of data transmitted at one time
$N_{L2data}$	Amount of L2ACK message at one time	*B*	Bandwidth for uplink channel
γ	Target SINR	$T_{ACKtimeout}$	Time set for ACK timeout
$T_{ACKtime}$	Time for receiving ACK message	$T_{beacons}$	Time for receiving beacons
$I_{period}$	Transmission period	σ(n)	Average backoff time of *n*th transmission attempts

**Table 3 sensors-23-05197-t003:** Simulation parameters.

Parameters	Value
Distance between IoT and AP	1 m∼15 m
Distance between APs	20 m
Cell coverage of an AP	radius 15 m (circle)
Number of APs	3 iAPs (triangle position)
Number of IoT devices	50, 100, 150, 200
Sensing Data	64 bytes
TCP/IP header	40 bytes
$N_{data}$	104 bytes
ACK	14 bytes
$N_{L2ack}$	54 bytes
Bandwidth, *B*	160 kHz
SINR, γ	40 dB
Maximum retransmission number, *m*	3
Transmit mode power of IoT devices, $P_{tx}$	0.45 W
Adaptive Transmit power of IoT devices, $P_{tx}^{adaptive}$	<0.45 W
Receive mode power of IoT devices, $P_{rx}$	0.15 W
Sleep mode power of IoT devices, $P_{sleep}$	0.00009 W
ACKtime	44 μs
ACKtimeout	337 μs
Transmission period, $I_{period}$	1 s
DTIM value, $n_{dtim}$	3
Beacon interval, $I_{beacon}$	100 ms
SIFS	10 μs
DIFS	50 μs
Slot time	20 μs
Initial contention window size, $CW_{min}$	16
Maximum contention window size, $CW_{max}$	128
Distribution ratio of the [$AP_{1}$:*AP*${}_{2}$:*AP*${}_{3}$]	[1:1:1], [1:9:9], [1:10:3]

**Table 4 sensors-23-05197-t004:** Training and Inference time of EL-RL model.

Distribution Ratio	1:1:1				1:9:9				1:10:3			
Number of IoT devices	50	100	150	200	50	100	150	200	50	100	150	200
Training time (s)	1.88	4.56	7.72	12.4	2.28	4.95	8.23	14.6	2.61	4.67	8.51	14.2
Inference time (ms)	0.255	0.279	0.293	0.275	0.265	0.271	0.239	0.299	0.264	0.235	0.288	0.272

## Data Availability

Not applicable.

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
