# Peer review of "Energy-Efficient AP Selection Using Intelligent Access Point System to Increase the Lifespan of IoT Devices"

_sensors, 2023, doi:10.3390/s23115197_

Round 1

Reviewer 1 Report

The paper proposed an energy-efficient AP selection scheme for IoT devices using reinforcement learning to reduce unnecessary packet transmission activities caused by collisions. The EL-RL model is utilized to achieve significant improvements in energy efficiency, uplink latency, and expected lifespan of IoT devices in crowded environments with multiple overlapping cells. Herein few comments which improve the quality of the article:

*Summarize the contributions of the article in 3-4 concise and focused points to enhance clarity and impact.

*Section 2 (Related Works), could be improved by summarizing the findings of each article in a table format, which would make it easier for readers to compare and contrast the different contributions. Additionally, the section could be updated to include more recent references, such as  (https://doi.org/10.3390/sym15030757);;(DOI:10.1109/MCOM.2017.1600218CM(https://doi.org/10.3390/sym12010088) 

*Increase the font size in the figures to improve readability and enhance the quality of the paper.

*While the results presented in the paper are satisfactory, we recommend improving the resolution of the figures to enhance their quality and clarity. This will ensure that the data and findings presented in the figures are easily interpretable and provide a strong visual representation of the research.

*It would be interesting to know more about the potential limitations and future directions of the proposed scheme, as well as how it might be adapted or extended for different types of IoT devices or environments.

The language used in the paper is generally readable, although there are some minor issues with typos and grammar. While these issues do not significantly detract from the overall quality of the paper, addressing them would improve its clarity and professionalism.

Reviewer 2 Report

This paper presents an energy-efficient AP selection scheme using reinforcement learning. It is interesting  and well-structure. The simuation results can support the author's claim. I have some comments.

(1)  In figure 5, AI models are installed in iAP, but figures 2 and 3 did not present it. Should AI models be installed in iAP?

(2) Where the iAP controller should be deployed in the proposed system?

(3) When the EL-RL traning model should be performed? How much time needed when EL-RL training model performs?

Reviewer 3 Report

This article presents an AP selection algorithm for enhancing the lifecycle of IoT devices. The article is well structured and supported by extensive experimental results. Some improvements still need to be made before it can be accepted:

1)The authors list 5 points as the main contributions of the article. I think this is too many. For example, points 4 and 5 are both analyses of experimental results, which I think could be combined;

2)Regarding AP selection in IoT environment, many application cases can be found, such as data collection of mobile objects in IoT environments. Regarding these related applications, the authors can reference “Fresh data collection for UAV-assisted IoTs based on proximity-remote region collaboration”; “Fresh Data Collection for UAV-Assisted IoTs based on Aerial Collaborative Relay”;

3)The authors propose to use a reinforcement learning algorithm for AP selection. The AP selection problem is not a very complex problem. For this problem, is the reinforcement learning algorithm too complex? The authors should fully argue in the introduct section why it is necessary to use such a complex reinforcement learning algorithm.

4)The authors should add a list of notations at the beginning to help the reader understand the proposed algorithm.

5)The authors mention that training is needed in the EL-RL model. The authors should give more explanation on how to train the data.

The language quality should be improved. 

Reviewer 4 Report

This submission presents a energy-efficient AP selection scheme using reinforcement learning to address the problem of unbalanced load that arises from biased AP connections. The proposed method utilizes the Energy and Latency Reinforcement Learning (EL-RL) model for energy-efficient AP selection that takes into account the average energy consumption and the average latency of IoT devices. In the EL-RL model, This submission analyzes the collision probability in Wi-Fi networks to reduce the number of retransmissions that induces more energy consumption and higher latency.

1.This submission has not sufficiently clarified the novelty of the proposed approach. 

2. This submission misses discussing a few relevant works, such as

- "Power saving for machine to machine communications in cellular networks", in Proc. IEEE Global Communications Conference 2011 Workshops

- “Big Data Meet Green Challenges: Greening Big Data”, IEEE Systems Journal, vol. 10, no. 3, Sept. 2016 

Round 2

Reviewer 1 Report

The article has been improved and can be published

good

Reviewer 3 Report

Can be Accepted.